# Association between neutrophil count and the risk of cardiovascular disease: A community-based cohort study in Taiwan

Kuang-Chung Wang[1], Chu-Lin Lin[2], Chun-Chieh Lin[2], Yun-Tzu Lee[3], Le-Yin Hsu[4,5], Kuo-Liong Chien[4,6,7], Tzu-Lin Yeh[3,8]*

1 Department of Family Medicine, National Taiwan University Hospital Hsin-Chu Branch, Hsinchu, Taiwan, 2 Department of General Internal Medicine, MacKay Memorial Hospital, Taipei, Taiwan, 3 Department of Medicine, MacKay Medical College, New Taipei, Taiwan, 4 Institute of Epidemiology and Preventive Medicine, College of Public Health, National Taiwan University, Taipei, Taiwan, 5 Graduate Program of Data Science, National Taiwan University and Academia Sinica, Taipei, Taiwan, 6 Department of Internal Medicine, National Taiwan University Hospital, Taipei, Taiwan, 7 Population Health Research Center, National Taiwan University, Taipei, Taiwan, 8 Department of Family Medicine. Hsinchu MacKay Memorial Hospital, Hsinchu, Taiwan

* 5767@mmh.org.tw

## Abstract

### Background

Neutrophil count is associated with atherosclerotic plaque formation and cardiovascular diseases (CVD). As previous studies have been predominantly conducted in Caucasians, the significance of neutrophil count as a clinical factor in CVD in other ethnicities remain unclear.

### Methods

A total of 2,955 participants from the Chin-Shan Community Cardiovascular Study(C-CCC), who had no established CVD diagnosis and missing data, were enrolled in this study and followed from 1990–1991–2013. We use Cox regression models to calculate hazard ratio (HR) and 95% confidence interval (CI) to evaluate the association between neutrophil count and CVD risk. Subgroup analyses were performed based on sex and age, while sensitivity analyses were conducted by excluding participants with extreme values.

### Results

Over a median follow-up period of 22 years, 400 cases of new-onset CVD were recorded. Cox proportional hazards regression analysis revealed that a higher neutrophil count was independently associated with CVD incidence in Taiwanese adults, with an HR of 1.42 (95% CI 1.03–1.94) after adjusting for multiple covariates. This association remained consistent in both the subgroup and sensitivity analyses.

**Data availability statement:** Data are available from the Taipei MacKay Memorial hospital ethics committee (contact via mmhirb82@gmail.com) for researchers who meet the criteria for access to confidential data.

**Funding:** The author(s) received no specific funding for this work.

**Competing interests:** The authors have declared that no competing interests exist.

## Conclusion

Our study demonstrated that, in the Taiwanese population, a higher neutrophil count was associated with a higher incidence of CVD over an average 22-year follow-up in individuals without preexisting CVD.

## Introduction

Cardiovascular disease (CVD) is the leading cause of mortality globally and accounts for approximately 18 million deaths annually, representing one-third of all deaths worldwide. Most fatalities (approximately 85%) are attributed to coronary artery disease (CAD) and cerebrovascular accidents (CVA) [1,2]. According to the 2022 national statistics in Taiwan, CAD and CVA rank second and fifth, respectively, among the top ten causes of death [3]. Established risk factors for CVD include sex, age, race, family history, hypertension, hypercholesterolemia, smoking, obesity, diabetes, poor diet, and physical inactivity [4–6]. A study by J. Fan indicates a concerning trend, with global ischemic stroke-related deaths rising from 2.04 million to 3.29 million between 1990 and 2019, and an estimated increase to 4.9 million by 2030 [7]. In addition to the risk factors mentioned above, the relationship between CVD and complete blood cell count (CBC), particularly the neutrophil count, has been increasingly emphasized [8–10].

Neutrophil count, which is calculated from a complete blood count, is an inexpensive and routine blood examination for the general population. Furthermore, the major CVD causes, such as atherosclerotic plaque formation and arterial thrombosis were associated with neutrophil extracellular traps expelled from suicidal neutrophils [11,12]. Therefore, we hypothesize that the increase in neutrophil count may be associated with the development of cardiovascular disease.

Previous studies have investigated the association between neutrophil count and CVD incidence. Most studies have shown the association between elevated neutrophil count and an increased CVD incidence [13–21] except a few studies with contradictory results [22,23].

As previous studies have been predominantly conducted on Caucasian participants, with some studies on Chinese and Japanese, the research on the Taiwanese population is lacking. Additionally, most studies have focused on analyzing white blood cell (WBC) counts and their differentials. Few studies have conducted detailed analyses of single differential counts. Therefore, we aimed to investigate whether higher neutrophil counts could predict incident CVD in the Taiwanese population.

## Methods

### Study population

We recruited 3,602 participants from the Chin-Shan Community Cardiovascular Study (CCCC) and conducted a retrospective cohort study. The CCCC is a cohort study aimed at reflecting the cardiovascular health status of the Taiwanese population through the demographic composition of residents in the Chin-Shan area. [24–26].

We collected the participants' underlying diseases and baseline health status, such as age, sex, smoking history and alcohol consumption, using standard questionnaires. Laboratory data, including complete blood counts and metabolic-related markers, were obtained by the same technicians at the same medical facility during 1990–1991. Participants aged > 35 years old who live in Chin-Shan area in Taiwan and signed the informed consent form were included; however, those who had missing laboratory data, CAD, and CVA before the study and those who were lost or unwilling to follow up were excluded from the study.

This study was approved by the Institutional Review Board of MacKay Memorial Hospital on Jan,22,2024 and conducted in accordance with the Declaration of Helsinki [27]. Data were accessed by the authors for research purposes on February 13, 2024. The authors did not have access to information that could identify individual participants during or after data collection.

To evaluate whether our sample size was sufficient to detect the observed association, we conducted a sample size estimation. We set the alpha level at 0.05 and the power (1−β) at 0.80, assuming a ratio of Neutrophil Q4 to Q1 of approximately 1:1. Based on previous literature indicating that a 10.6% incidence of CVD in Q1 and a relative risk (RR) of 1.57 for Q4 compared with Q1[13], the required sample size was estimated to be 1,380 participants under these assumptions.

Participants included in this study were categorized into neutrophil quartiles. Other blood cell counts, including red blood cells (RBC), WBC, lymphocytes, and platelets, were also categorized.

The endpoint of the study was the incidence of cardiovascular events ascertained from hospital records or death certificates, and participants were followed-up until the end of 2013. Cardiovascular events included those related to CAD and CVA. The definition of CAD included fatal and nonfatal CADs, confirmed through death certificate hospitalization records of conditions that required coronary artery bypass grafting (CABG) or coronary angioplasty, respectively. The definition of CVA included neurological symptoms of vascular origin lasting more than 24 hours, resulting in hospitalization or death. CVA subtypes, including hemorrhagic and ischemic strokes, were diagnosed using imaging; however, patients with transient ischemic attacks (TIAs) were excluded. For individuals who experienced both types of events, the endpoint was defined as the time of the first event.

Covariates were derived from questionnaires and hematology tests of the CCCC from to 1990–1991. The chosen categorical and continuous variables included sex and age and fasting glucose, total cholesterol, triglycerides, high-density lipoprotein (HDL), low-density lipoprotein (LDL), systolic blood pressure, and body mass index (BMI), respectively[4–6,28,29]. (S1 Table)

All participants were classified into four groups according to the quartiles of each blood cell count. The covariates of each group (sex, demographic distribution, and clinical variables) were presented as baseline characteristics. The results for continuous variables were presented as means with standard deviations; variables among the four groups were compared using Analysis of Variance (ANOVA) [30]. The results for categorical variables are presented as frequencies with percentages, and the chi-square test was used to compare the distributional differences among the four groups [31].

This study used Kaplan–Meier survival curves to analyze the survival rate according to the study endpoint and the log-rank test to test the statistical significance among the survival curves of each group [32,33]. The calculation of person-years began with the date of the first interview (1990–1991) and ended on December 31, 2013, the date of cardiovascular events or death diagnosis, whichever came first, during the study period. The incidence rate in this study was defined as the number of events per 1,000 person-years.

We used Cox proportional hazard regression models to calculate hazard ratios (HR) and 95% confidence intervals (CI) for outcomes linked to neutrophil count [34]. The proportional hazard assumption of proportional hazards, evaluated by introducing a product term for follow-up time and neutrophil count, revealed no significant deviation from the assumption. Cox proportional hazards regression analysis was used to analyze the data. Three multivariable analysis models were constructed: Model 1 was adjusted for age and sex; Model 2 incorporated additional lifestyle factors such as BMI, smoking, and alcohol consumption; and Model 3 further incorporated clinical variables such as systolic blood pressure, fasting blood glucose, total cholesterol, HDL, and LDL.

## Subgroup analysis

We examined potential effect modifiers, specifically sex and age (with a cut-off of 65 years), using the likelihood ratio test. This test was used to compare the goodness-of-fit of the models with and without interaction terms within the fully adjusted model.

## Sensitivity analysis

We conducted sensitivity analyses by excluding participants at higher risk of stroke or thrombosis, such as those with higher hemoglobin levels (>16.5 g/dL), and abnormal platelet counts (>450 x $10^3$ or <100 x $10^3$/μL).

All statistics were two-tailed tests with a significance level of $p < 0.05$. Statistical significance was set at $p < 0.05$. 3. SAS software version 9.4 (TS Level 1M7) and Stata version 16.1 (Stata Corporation, College Station, TX 77845, USA) were used for the analysis.

## Results

Of the 3,602 adults enrolled in the CCCC study, after excluding the 648 because of established CAD (n = 173), established CVA (n = 84), missing data (n = 428), and outcome data that occurred before the index data or after the death date (n = 1), 2,955 individuals were enrolled in this study. (S1 Fig).

The baseline characteristics of the participants according to neutrophil count are listed in Table 1. The mean (standard deviation) age of the participants was 54.2 (12.2); 53.5% were women, and the mean BMI was 23.5 kg/m². In addition, 30.9% of the participants were smokers and 23.9% used alcohol. Participants were equally classified into four groups based on the distribution of neutrophil count (Q1: 0.7–3.0, Q2: 3.0–3.8, Q3: 3.8–4.7, Q4: >4.7–41.3 x $10^3$/mL). Individuals with higher neutrophil levels were more likely to be men, current smokers, drinkers, or obese. High neutrophil levels were significantly associated with elevated levels of systolic blood pressure, diastolic blood pressure, fasting plasma glucose,

**Table 1. Baseline characteristics of participants by neutrophil count.**

| Characteristics | Total | Neutrophil count | | | | p value |
|---|---|---|---|---|---|---|
| | | Q1 | Q2 | Q3 | Q4 | |
| | | 0.7–3.0 (10³/uL) | 3.0–3.8 (10³/uL) | 3.8–4.7 (10³/uL) | 4.7–41.3 (10³/uL) | |
| | n(%) | n(%) | n(%) | n(%) | n(%) | |
| **Age** | | | | | | 0.17 |
| 35–64 years old | 2,318 (78.4) | 524 (81.0) | 589 (76.2) | 580 (78.0) | 625 (79.0) | |
| ≥65 years old | 637 (21.6) | 123 (19.0) | 184 (23.8) | 164 (22.0) | 166 (21.0) | |
| **Sex** | | | | | | |
| Woman | 1,581 (53.5) | 411 (63.5) | 436 (56.4) | 386 (51.9) | 348 (44) | <0.001 |
| Current smoker | 913 (30.9) | 143 (22.1) | 221 (28.6) | 241 (32.4) | 308 (38.9) | <0.001 |
| Alcohol use | 703 (23.8) | 132 (20.4) | 183 (23.7) | 178 (23.9) | 210 (26.6) | 0.06 |
| | mean±SD | mean±SD | mean±SD | mean±SD | mean±SD | |
| Body mass index (kg/m²) | 23.5±3.4 | 22.8±3.3 | 23.3±3.2 | 23.8±3.6 | 23.8±3.5 | <0.001 |
| Systolic blood pressure (mmHg) | 125±20.2 | 123±18.9 | 124.7±19.8 | 126.3±20.5 | 125.7±21 | 0.012 |
| Diastolic blood pressure (mmHg) | 77±11.1 | 75.9±10.6 | 76.7±10.9 | 78±11.4 | 77.4±11.2 | 0.003 |
| Fasting plasma glucose (mg/dL) | 109.8±31.3 | 106.6±24.1 | 108.4±29.2 | 112.1±36.3 | 111.6±33.3 | 0.002 |
| Total cholesterol (mg/dL) | 196.8±44.6 | 193.1±45.4 | 194.9±44.1 | 196.2±42.4 | 202.4±46.2 | <0.001 |
| Triglycerides (mg/dL) | 125.1±94.9 | 106.2±80.4 | 117.9±89.4 | 134.2±104.2 | 139±98.7 | <0.001 |
| High-density lipoprotein cholesterol (mg/dL) | 47.6±12.4 | 49.8±12.6 | 48.3±12.4 | 45.7±11.4 | 47±12.9 | <0.001 |
| Low-density lipoprotein cholesterol (mg/dL) | 137±43.5 | 131.6±44.2 | 134.7±42.6 | 137.8±42 | 142.9±44.5 | <0.001 |

total cholesterol, triglycerides, and LDL, and low HDL levels. S2–S5 Table presents the detailed baseline characteristics of the participants by counts of RBC, WBC, and lymphocyte.

We used the first CAD or CVA event to calculate the person-years if a participant had experienced both CAD and CVA events. The Kaplan–Meier survival curves for CVD are shown in Fig 1. During a median follow-up period of 22 years, participants with a higher neutrophil count had a significantly lower survival rate than those with a lower neutrophil count (log-rank test, p<0.001). Survival curves for the analysis of CVD by RBC, WBC, lymphocyte, and platelet counts are plotted in S2–S5 Figs.

Table 2 shows the enrollment of 2,955 participants, contributing to 52,325 person-years. A total of 400 incident cases were recorded, with a median follow-up of 22.36 years (interquartile range, 10.66 years). The incidence of CVD was 7.64 per thousand person-years. Table 2 shows the results of the multivariable Cox regression models, indicating that the incidence of cardiovascular disease significantly increased with an increase in neutrophil count. In model 1, the HRs (95% CI) of CVD for participants in the Q4 group were 1.76 (1.29, 2.40) compared with participants in the Q1 group (*p* for trend<0.001). In model 2, the HRs (95% CI) of CVD for participants in the Q4 group were 1.63 (1.19, 2.22)(*p* for trend <0.05). In model 3, the HRs (95% CI) of CVD for participants in the Q4 group were 1.42 (1.03, 1.94) (*p* for trend<0.05). The results of all other blood cell count analyses are shown in S6-9 Tables.

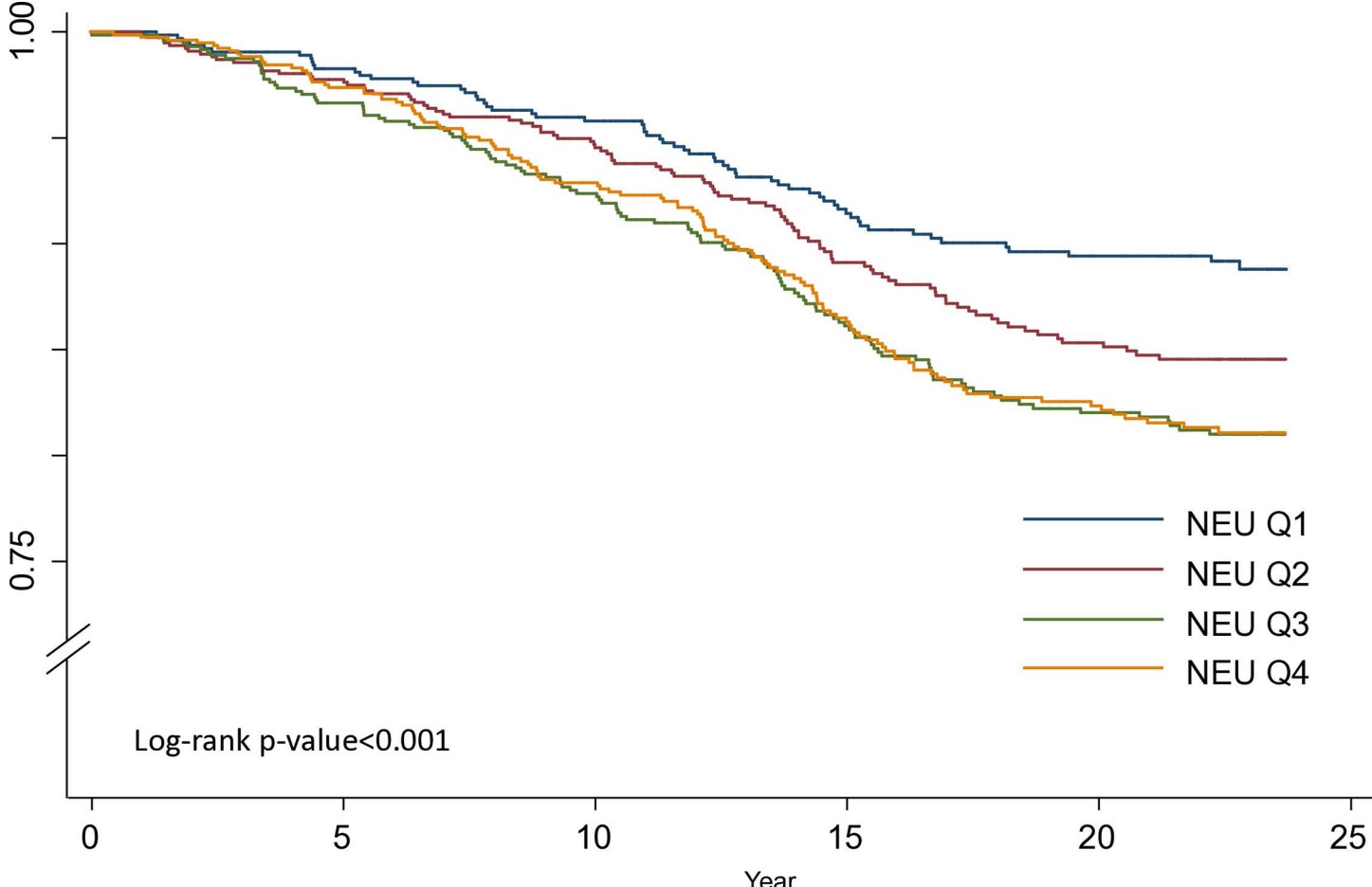

**Fig 1. The Kaplan-Meier survival curves for cardiovascular disease of neutrophil count.**

A subgroup analysis was conducted according to stratification by age and sex. As shown in Table 3, age and sex were not the effect modifiers for CVD risk. The interaction between sex, neutrophil count, and CVD risk was not significant ($p = 0.59$). The HR with 95% CI of the Q4 group in men and women was 1.69 (1.06, 2.69), and 1.25 (0.80, 1.96), respectively ($p = 0.003$). Similarly, no significant interaction was observed between age and neutrophil count on CVD risk ($p = 0.18$). The HR with 95% CI of the Q4 group in the participants aged <65 years was 1.54 (1.03, 2.31) and 1.12 (0.67, 1.87) in participants aged ≥65 years. The results of all analyses of other blood cell counts stratified by age and sex are shown in S10–13 Tables.

The results of the sensitivity analyses are presented in Table 4. We observed that after excluding extreme values of hemoglobin, the HR for Q4 compared to Q1 was 1.42 (1.02, 1.96), with a significant p-value for trend (p < 0.05).

**Table 2. Cardiovascular disease incidence according to the quartiles of neutrophil count.**

| Variables | Q1 | Q2 | Q3 | Q4 | |
|---|---|---|---|---|---|
| Participants | 647 | 773 | 744 | 791 | |
| Person-years | 12,105 | 13,819 | 12,894 | 13,507 | |
| Events | 62 | 99 | 118 | 121 | |
| Incidence rate per 1000-person years | 5.12 | 7.16 | 9.15 | 8.96 | |
| **Hazard ratio (95% CI)** | | | | | **p for trend** |
| | Ref. | 1.40 (1.02-1.93) | 1.79 (1.32-2.43) | 1.76 (1.29-2.39) | <0.001 |
| Model 1 | Ref. | 1.31 (0.95-1.80) | 1.68 (1.24-2.29) | 1.76 (1.29-2.40) | <0.001 |
| Model 2 | Ref. | 1.25 (0.91-1.72) | 1.53 (1.12-2.09) | 1.63 (1.19-2.22) | 0.001 |
| Model 3 | Ref. | 1.18 (0.86-1.63) | 1.40 (1.02-1.91) | 1.42 (1.03-1.94) | 0.026 |

Model 1: adjusted for age and sex; Model 2: adjusted for model 1, body mass index, current smoking, alcohol use; Model 3: adjusted for model 2, systolic blood pressure, fasting plasma glucose, total cholesterol, high-density lipoprotein, and low-density lipoprotein.

Abbreviations: CI, confidence interval

**Table 3. Subgroup analysis of the cardiovascular disease incidence according to the quartiles of neutrophil count.**

| Variables | Q1 | Q2 | Q3 | Q4 | p-value for interaction |
|---|---|---|---|---|---|
| Age | | | | | 0.18 |
| 35–64 years old | 1 | 1.29 (0.84-1.97) | 1.71 (1.14-2.57) | 1.54 (1.03-2.31) | |
| ≥65 years old | 1 | 1.03 (0.63-1.69) | 1.04 (0.63-1.72) | 1.12 (0.67-1.87) | |
| Sex | | | | | 0.59 |
| Men | 1 | 1.31 (0.80-2.14) | 1.79 (1.11-2.87) | 1.69 (1.06-2.69) | |
| Women | 1 | 1.11 (0.73-1.69) | 1.13 (0.74-1.73) | 1.25 (0.80-1.96) | |

Hazards ratio was adjusted using Model 3 (age, sex, body mass index, current smoking, alcohol use, systolic blood pressure, fasting plasma glucose, total cholesterol, high-density lipoprotein, and low-density lipoprotein).

**Table 4. Sensitivity analysis of the cardiovascular disease incidence according to the quartiles of neutrophil count.**

| | Neutrophil count | | | | |
|---|---|---|---|---|---|
| Variables | Q1 | Q2 | Q3 | Q4 | p-value for trend |
| Exclude extreme data[a] | Ref. | 1.20 (0.87-1.66) | 1.31 (0.95-1.81) | 1.42 (1.02-1.96) | 0.035 |
| Exclude extreme data[b] | Ref. | 1.14 (0.83-1.56) | 1.37 (1.02-1.84) | 1.33 (0.98-1.80) | 0.05 |

a: Extreme data include: Hb > 16.5 g/dL

b: Extreme data include platelet> 450 x 10³/µL or < 100 x 10³/µL

Conversely, after excluding extreme values of platelets, the HR for Q4 compared to Q1 was 1.33 (0.98–1.80), with a p-value for trend of 0.05. The results of the sensitivity analysis were consistent with the main results. The results of all analyses of the other blood cell counts, excluding extreme values, are shown in S14–17 Tables.

## Discussion

This retrospective cohort study with an average follow-up of 22 years demonstrated that a higher neutrophil count was independently associated with CVD incidence in Taiwanese adults aged 35 and above.As shown in Table 2, we observed a clear dose–response trend, indicating that as neutrophil count increased, the risk of CVD also rose. In Model 1, participants in Q4 had a hazard ratio (HR) of 1.76 (95% CI 1.29–2.40), corresponding to a 76% higher hazard of CVD compared with Q1. Even in the fully adjusted Model 3, Q4 retained a significantly elevated risk (HR 1.42, 95% CI 1.03–1.94)—that is, a 42% higher hazard relative to Q1. Notably, these patterns were consistent in the subgroup analysis (Table 3), where stratification by age and sex demonstrated that this positive association persisted across different demographic groups. Furthermore, the sensitivity analyses (Table 4), which excluded participants with extreme data points (e.g., Hb > 16.5 g/dL or platelets >450 × 10^3/μL or < 100 × 10^3/μL), produced comparable results, further underscoring the robustness of our findings.

The results of other blood cell counts analyses showed a significance in the WBC group after adjusting for multiple covariates. The lymphocyte and platelet groups showed no significant association between blood cell count and CVD incidence.

A review of similar studies showed that most support an association between neutrophil count and the incidence of CVD [13,15–17,19,21,35]. In the Atherosclerosis Risk in Communities study, an elevated neutrophil count was associated with an increased incidence of coronary heart disease, ischemic stroke, and mortality in African American and Caucasian men and women [35].. Furthermore, in a study using Copenhagen General Population Study (CGPS) and the UK Biobank, individuals(n = 563,085) with higher neutrophil counts were associated with ischemic heart disease and peripheral arterial disease[17]. However, most studies primarily analyzed and compared WBC and differential counts simultaneously, with few studies specifically analyzing neutrophils.

The potential mechanisms linking neutrophil counts to the risk of CVA and CHD are currently being explored. Neutrophils play a critical role in the onset and progression of atherosclerosis and CVD [36] as they are the first blood cells to be recruited to sites of arterial damage, inducing strong inflammatory responses and exacerbating endothelial damage [37,38]. Neutrophils secrete reactive oxygen species and proteases, contributing to necrotic core formation and release of chemotactic proteins that promote macrophage expansion [36,39]. They stimulate macrophages, promote foam cell formation, and accelerate their transformation into foam cells [36]. Neutrophils and their extracellular traps also play a significant role in priming and activating inflammatory macrophages, as well as in stimulating the initiation of B cells and T cell activity, leading to interleukins and interferon-alpha release [40–42].

Neutrophils are also crucial in plaque instability, including plaque rupture and plaque erosion [43,44]. During plaque rupture, neutrophils expand the necrotic core, disrupt the arterial wall, and promote fibrous cap thinning and collagen degradation [45]. Contrarily, during plaque erosion, neutrophils cause endothelial cell death and detachment [46,47] and promote platelet activation and coagulation, leading to thrombus formation [38,48].

Based on these results, the neutrophil group was the most predictive blood cell type for the occurrence of CVD compared to the WBC, RBC, lymphocyte, and platelet groups. This finding was consistent regardless of the age or sex. Unlike other CVD screening methods such as calcium score, coronary computed tomography angiography, or carotid ultrasound, neutrophil count is easily obtainable and inexpensive. Although it does not directly indicate the disease severity, it can inform patients that the neutrophil count correlates with CVD risk. This can lead to the further examination, diagnosis, treatment, and education of patients on improving their lifestyle habits to reduce cardiovascular risk.

Our study had some limitations. Neutrophil counts can be influenced by current infections, physiological stress, and medication use, which may introduce variability into the measurements. While we employed quartile grouping to mitigate the influence of extreme values [49–51]. Additionally, the neutrophil counts were measured only once at at

the time of blood collection, which may not accurately reflect long-term trends or variations over the 22-year follow-up period. This limitation could potentially obscure the dynamic relationship between neutrophil levels and cardiovascular disease risk. Third, while our study identified a significant association between elevated neutrophil counts and increased CVD risk, the clinical significance of these findings remains unclear. Further research is needed to determine how individuals with higher neutrophil counts can effectively modify their lifestyle to lower their risk [52].

## Conclusion

Our study demonstrated that, in the Taiwanese population, a higher neutrophil count was associated with a higher incidence of CVD over an average 22-year follow-up in individuals without preexisting CVD.

## Supporting information

**S1 Table. Operational definition of covariate.**
(DOCX)

**S2 Table. Baseline characteristics of participants by red blood cell.**
(DOCX)

**S3 Table. Baseline characteristics of participants by white blood cell.**
(DOCX)

**S4 Table. Baseline characteristics of participants by lymphocyte count.**
(DOCX)

**S5 Table. Baseline characteristics of participants by platelet.**
(DOCX)

**S6 Table. The cardiovascular disease incidence according to the quartiles of red blood cell.**
(DOCX)

**S7 Table. The cardiovascular disease incidence according to the quartiles of white blood cell.**
(DOCX)

**S8 Table. The cardiovascular disease incidence according to the quartiles of lymphocyte count.**
(DOCX)

**S9 Table. The cardiovascular disease incidence according to the quartiles of platelet.**
(DOCX)

**S10 Table. Subgroup analysis of the cardiovascular disease incidence according to the quartiles of red blood cell.**
(DOCX)

**S11 Table. Subgroup analysis of the cardiovascular disease incidence according to the quartiles of white blood cell.**
(DOCX)

**S12 Table. Subgroup analysis of the cardiovascular disease incidence according to the quartiles of lymphocyte count.**
(DOCX)

**S13 Table. Subgroup analysis of the cardiovascular disease incidence according to the quartiles of platelet.**
(DOCX)

**S14 Table. Sensitivity analysis of the cardiovascular disease incidence according to the quartiles of red blood cell.**
(DOCX)

**S15 Table. Sensitivity analysis of the cardiovascular disease incidence according to the quartiles of white blood cell.**
(DOCX)

**S16 Table. Sensitivity analysis of the cardiovascular disease incidence according to the quartiles of lymphocyte count.**
(DOCX)

**S17 Table. Sensitivity analysis of the cardiovascular disease incidence according to the quartiles of platelet.**
(DOCX)

**S1 Fig. Flow chart of study population selection.**
(DOCX)

**S2 Fig. The Kaplan-Meier survival curves for cardiovascular disease of RBC.**
(DOCX)

**S3 Fig. The Kaplan-Meier survival curves for cardiovascular disease of WBC.**
(DOCX)

**S4 Fig. The Kaplan-Meier survival curves for cardiovascular disease of lymphocyte.**
(DOCX)

**S5 Fig. The Kaplan-Meier survival curves for cardiovascular disease of platelet.**
(DOCX)

## Acknowledgments

We would like to express our sincere gratitude to Professor Chien for providing the database used in this study. Special thanks to Hsien-Yu Fan for his assistance with this journal.

## Author contributions

**Conceptualization:** Kuang-Chung Wang, Chu-Lin Lin, Chun-Chieh Lin, Kuo-Liong Chien, Tzu-Lin Yeh.

**Data curation:** Kuang-Chung Wang, Chun-Chieh Lin, Le-Yin Hsu, Kuo-Liong Chien.

**Formal analysis:** Kuang-Chung Wang.

**Investigation:** Kuang-Chung Wang, Tzu-Lin Yeh.

**Methodology:** Kuang-Chung Wang, Chu-Lin Lin, Yun-Tzu Lee.

**Resources:** Kuang-Chung Wang, Tzu-Lin Yeh.

**Supervision:** Le-Yin Hsu, Kuo-Liong Chien, Tzu-Lin Yeh.

**Writing – original draft:** Kuang-Chung Wang, Chun-Chieh Lin, Yun-Tzu Lee.

**Writing – review & editing:** Kuang-Chung Wang, Tzu-Lin Yeh.

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
