## [Decision Letter · Decision Letter 0]

29 Dec 2024

PONE-D-24-42923Association between neutrophil count and the risk of cardiovascular disease: A community-based cohort study in TaiwanPLOS ONE

Dear Dr. Yeh,

Thank you for submitting your manuscript to PLOS ONE. After careful consideration, we feel that it has merit but does not fully meet PLOS ONE’s publication criteria as it currently stands. Therefore, we invite you to submit a revised version of the manuscript that addresses the points raised during the review process.

We look forward to receiving your revised manuscript.

Kind regards,

Aleksandra Klisic

Academic Editor

PLOS ONE

Journal Requirements:

2. In the online submission form, you indicated that the datasets generated and analyzed during the current study are not publicly available due to the terms of consent to which the participants agreed but data are however available from the authors upon reasonable request.

Reviewers' comments:

Reviewer's Responses to Questions

**Comments to the Author**

1. Is the manuscript technically sound, and do the data support the conclusions?

Reviewer #1: Yes

Reviewer #2: Partly

Reviewer #3: Yes

2. Has the statistical analysis been performed appropriately and rigorously? 

Reviewer #1: Yes

Reviewer #2: Yes

Reviewer #3: Yes

3. Have the authors made all data underlying the findings in their manuscript fully available?

Reviewer #1: Yes

Reviewer #2: Yes

Reviewer #3: Yes

4. Is the manuscript presented in an intelligible fashion and written in standard English?

Reviewer #1: Yes

Reviewer #2: Yes

Reviewer #3: Yes

5. Review Comments to the Author

Reviewer #1: This manuscript investigates the association between neutrophil count and cardiovascular disease (CVD) risk in a Taiwanese population using a retrospective cohort study with a 22-year follow-up period. The study analyzes data from 2,955 participants of the Chin-Shan Community Cardiovascular Study. It identifies a significant association between elevated neutrophil counts and increased CVD incidence, with a hazard ratio of 1.76 for participants in the highest quartile of neutrophil count compared to the lowest.

I would like to commend the authors for conducting this valuable study. The methodology and statistical analyses are well-structured and performed. However, several points require attention to enhance the overall quality and clarity of the manuscript:

1. To strengthen the validity of your findings, consider including a sample size estimation or a post-hoc power calculation for the primary outcome. This addition will provide insight into the study's statistical power and robustness.

2. It would be beneficial to clarify whether certain factors in your analysis were treated as covariates or confounders. Reviewing the existing literature to appropriately classify these factors and using precise terminology will improve the study's methodological rigor. Although the overall analysis is correct.

3. Given the presence of multiple comparisons/testing and stepwise analyses in the study, it is crucial to adjust the standard significance level (e.g., using Bonferroni correction) to mitigate the risk of Type I errors. Such adjustments might also impact the reported confidence intervals and should be considered.

4. On page 10 (line 203), the term "multivariate Cox regression models" is used. The appropriate term in this context appears to be "multivariable," as it more accurately reflects the inclusion of multiple independent variables rather than dependent ones.

5. Including interpretation zones and cut-offs for hazard ratios (HR), supported by references, will enrich the methods and discussion sections. This approach can help contextualize the findings more effectively for the audience.

6. The dose-response trend in HR values observed in Table 2 is noteworthy. Highlighting this trend in your interpretation will add depth to the results and strengthen the discussion.

7. The discussion section would benefit from using simple words for readers. For instance, rather than merely stating an HR of 1.72, explain that this translates to a 72% higher hazard in group 4 compared to group 1. Such simplifications will enhance reader comprehension.

Reviewer #2: Hello

It is a good article, but almost the same article was done in 2023, which is available on the Internet. Therefore, your article should investigate the topic further or the discussion and conclusion section should further investigate and explain the mechanism of connection and correlation. In my opinion, make the discussion and conclusion section more complete.

Good luck.

Reviewer #3: REVIEWER COMMENT AND SUGGESTION

I would like to congratulate the Authors for raising the important topic pertaining the patient cantered care. When, I have been reading throughout the introduction and discussion part was really exciting and interesting. Hope this paper will contribute to new body of knowledge by introducing the Association between neutrophil count and the risk of cardiovascular disease.

•However, there are few comments and suggestions for authors for the sake of improving this interesting paper.

•Work extensively to be clear grammar and typographical errors throughout the document

ABSTRACT

•The authors should revise background I noted there are information is mixing which are not needed to this part. Line 30-40

Conclusion

•I comment the authors should conclude according to the result. Revise and improve

Introductions

•The authors should explain the scientific background and rationale for the investigation being reported. Improve

•I noted line 90 (CCCC) what does it meaning

•State specific objectives, including any prespecified hypotheses

METHOD

•I would ask the authors its retrospective design or prospective design. I comment revise and make it clear

•I noted line 124 the authors should correct it

Participant

•Give the eligibility criteria, and the sources and methods of selection of participants. Describe methods of follow-up

Data source

•For each variable of interest, give sources of data and details of methods of assessment (measurement). Describe comparability of assessment methods if there is more than one group

Bias

•Explain how the study size was arrived at

Result

• Give unadjusted estimates and, if applicable, confounder-adjusted estimates and their precision (eg, 95% confidence interval). Make clear which confounders were adjusted for and why they were included

• Report category boundaries when continuous variables were categorized

• If relevant, consider translating estimates of relative risk into absolute risk for a meaningful time period

Discussion

•Summaries key results with reference to study objectives

•Improve limitation

6. PLOS authors have the option to publish the peer review history of their article (what does this mean? ). If published, this will include your full peer review and any attached files.

**Do you want your identity to be public for this peer review?** For information about this choice, including consent withdrawal, please see our Privacy Policy .

Reviewer #1: **Yes: ** Mirsaeed Abdollahi

Reviewer #2: No

Reviewer #3: **Yes: ** rehema abdallah

---

## [Author Response · Author response to Decision Letter 1]

5 Feb 2025

Response to Reviewer #1

Dear Reviewer,

Thank you for your thoughtful comments and valuable suggestions, which have greatly contributed to improving the clarity and quality of our manuscript. Below, we address each of your comments in detail and outline the revisions made to the manuscript:

1. To strengthen the validity of your findings, consider including a sample size estimation or a post-hoc power calculation for the primary outcome. This addition will provide insight into the study's statistical power and robustness.

Response: Thank you for the suggestion. I have added a sample size estimation in the Methods section. We set the alpha level at 0.05 and the power (1−β) at 0.80, assuming a ratio of Neutrophil Q4 to Q1 of approximately 1:1. Based on previous literature, 10.6% incidence of CVD in Q1, and a relative risk (RR) of 1.57 for Q4 compared with Q1(ref.). Under these assumptions, our statistical software indicated that a total of 1,380 participants would be required.

Since our study included 2,955 participants—substantially higher than this estimated requirement—it is likely that we had adequate statistical power to detect the main effect of interest. This supplementary analysis therefore strengthens our primary findings and underscores the robustness of our conclusions.

Ref: Karino, S., et al., Total and differential white blood cell counts predict eight-year incident coronary heart disease in elderly Japanese-American men: the Honolulu Heart Program. Atherosclerosis, 2015. 238(2): p. 153-8.

2. It would be beneficial to clarify whether certain factors in your analysis were treated as covariates or confounders. Reviewing the existing literature to appropriately classify these factors and using precise terminology will improve the study's methodological rigor. Although the overall analysis is correct.

Response: Thank you for the suggestion. I have added references for the covariates used in this study at lines 134–137 to ensure clarity and to provide appropriate citations for the variables included in the analysis.

3. Given the presence of multiple comparisons/testing and stepwise analyses in the study, it is crucial to adjust the standard significance level (e.g., using Bonferroni correction) to mitigate the risk of Type I errors. Such adjustments might also impact the reported confidence intervals and should be considered.

Response: Thank you for your suggestion. In Table 2, we did not use ANOVA for our comparisons; rather, we used p-value for trend analysis to assess whether there is a linear increase in CVD risk as neutrophil count rises. This approach avoids the multiple comparisons issue that ANOVA can sometimes pose, and thus does not require a correction for multiple testing.

4.On page 10 (line 203), the term "multivariate Cox regression models" is used. The appropriate term in this context appears to be "multivariable," as it more accurately reflects the inclusion of multiple independent variables rather than dependent ones.

Response: Thank you for the reviewer’s suggestion. I have replaced "multivariate" with "multivariable" in the manuscript to ensure the terminology is accurate and appropriate.

5–7.

• Including interpretation zones and cut-offs for hazard ratios (HR), supported by references, will enrich the methods and discussion sections. This approach can help contextualize the findings more effectively for the audience.

• The dose-response trend in HR values observed in Table 2 is noteworthy. Highlighting this trend in your interpretation will add depth to the results and strengthen the discussion.

• The discussion section would benefit from using simple words for readers. For instance, rather than merely stating an HR of 1.72, explain that this translates to a 72% higher hazard in group 4 compared to group 1. Such simplifications will enhance reader comprehension.

Response: Thank you for the reviewer’s suggestions. I have revised the first paragraph of the “Discussion” to provide a more detailed explanation of the significance of increasing HR values and to elaborate on the dose–response trend observed in Table 2. Additionally, I have simplified the interpretation of the results to improve reader comprehension, such as explaining that an HR of 1.72 corresponds to a 72% higher hazard in group 4 compared to group 1.

Response to Reviewer #2:

Dear Reviewer,

Thank you for your thoughtful comments and valuable suggestions, which have significantly helped improve the quality of our manuscript. Below, we address each of your comments in detail and outline the revisions made to the manuscript.

Reviewer Comment: It is a good article, but almost the same article was done in 2023, which is available on the Internet. Therefore, your article should investigate the topic further or the discussion and conclusion section should further investigate and explain the mechanism of connection and correlation. In my opinion, make the discussion and conclusion section more complete.

Good luck.

Response: Thank you for your insightful suggestion. I have reviewed the 2023 article on the topic of neutrophil counts and cardiovascular disease and incorporated its relevant findings into the discussion. While the topic and main findings are consistent with our study, our research focuses on a cohort of Chinese adults followed over an extended 20-year period. This long-term follow-up allowed us to capture the natural history of cardiovascular disease in a stable environment, providing robust observational insights that complement genetic instrument-based approaches.

Response to Reviewer #3

Dear Reviewer,

Thank you for your thoughtful comments and valuable suggestions, which have significantly helped improve the quality of our manuscript. Below, we address each of your comments in detail and outline the revisions made to the manuscript.

1.Reviewer Comment: I would like to congratulate the authors for raising the important topic pertaining to patient-centered care. When reading the introduction and discussion parts, I found them exciting and interesting. I hope this paper will contribute to new knowledge by introducing the association between neutrophil count and the risk of cardiovascular disease.

Response: Thank you for your kind words and positive feedback on our study. We are encouraged by your comments and have addressed the specific issues you raised below.

2.Reviewer Comment: Work extensively to clear grammar and typographical errors throughout the document.

Response: Thank you for the suggestion. I have carefully reviewed the entire document and made extensive revisions to address grammatical issues and typographical errors. These changes ensure that the manuscript is more precise and reader-friendly while maintaining the integrity of the original findings.

3. Abstract

Reviewer Comment:

• Revise the background, as I noted information is mixing and not needed in this part (Line 30-40).

• Revise the conclusion to ensure it aligns with the results.

Response: Thank you for the suggestion. I have revised the abstract based on the study’s conclusions to improve clarity and ensure smoother phrasing.

4. Introduction

Reviewer Comment:

• The authors should explain the scientific background and rationale for the investigation being reported.

• Line 90: What does "CCCC" mean?

• State specific objectives, including any prespecified hypotheses.

Response: Thank you for the suggestion. I have revised the introduction to more clearly explain the rationale for the investigation and the hypothesis. Additionally, the database used in this study, derived from the Chin-Shan Community Cardiovascular Study (CCCC), has been described in greater detail in the Methods section.

5. Methods

(1) Reviewer Comment:

• Clarify whether this is a retrospective or prospective study.

• Line 124 needs to be revised for clarity.

Response: Thank you for the suggestion. I have added a statement in the manuscript clarifying that this study is a retrospective study. Additionally, I have revised Line 131-132(new location) to make the sentence clearer.

(2) Reviewer Comment:

• Provide the eligibility criteria, and the sources and methods used for selecting participants. Describe methods of follow-up.

Response: Thank you for the suggestion. I have revised the Methods section to provide a clearer description of the eligibility criteria, as well as the sources and methods used for selecting participants.

(3) Reviewer Comment:

• For each variable of interest, give sources of data and details of methods of assessment (measurement). Describe comparability of assessment methods if there is more than one group.

Response: Thank you for the suggestion. I have revised the Methods section to provide a clearer description of the participants’ underlying diseases and baseline health status, including age, sex, smoking history, and alcohol consumption, which were collected using standardized questionnaires. Laboratory data, including complete blood counts and metabolic-related markers, were obtained by the same technicians at the same medical facility during the 1990–1991 baseline period.

(4) Reviewer Comment:

• Explain how the study size was determined.

Response: Thank you for the suggestion. I have added a sample size estimation in the Methods section. We set the alpha level at 0.05 and the power (1−β) at 0.80, assuming a ratio of Neutrophil Q4 to Q1 of approximately 1:1. Based on previous literature, 10.6% incidence of CVD in Q1, and a relative risk (RR) of 1.57 for Q4 compared with Q1(ref.). Under these assumptions, our statistical software indicated that a total of 1,380 participants would be required.

Since our study included 2,955 participants—substantially higher than this estimated requirement—it is likely that we had adequate statistical power to detect the main effect of interest. This supplementary analysis therefore strengthens our primary findings and underscores the robustness of our conclusions.

Ref: Karino, S., et al., Total and differential white blood cell counts predict eight-year incident coronary heart disease in elderly Japanese-American men: the Honolulu Heart Program. Atherosclerosis, 2015. 238(2): p. 153-8.

6. Results

Reviewer Comment:

• Provide unadjusted and confounder-adjusted estimates and their precision (e.g., 95% confidence interval). Clearly indicate which confounders were adjusted for and why they were included.

• Report category boundaries when continuous variables were categorized.

• If relevant, consider translating estimates of relative risk into absolute risk for a meaningful time period.

Response: Thank you for the suggestion. I have added the unadjusted hazard ratio to Table 2. In addition, the neutrophil count category boundaries are as follows: Q1 (0.7–3.0), Q2 (3.0–3.8), Q3 (3.8–4.7), and Q4 (>4.7–41.3 × 10³/mL). These boundaries are provided in the second paragraph of the Results section and in Table 1.

Furthermore, regarding absolute risk, Table 2 already presents the incidence rates per 1,000 person-years for Q1 to Q4, which should effectively convey the overall risk to the readers.

7. Discussion

Reviewer Comment:

• Summarize key results with reference to study objectives.

• Improve the discussion of limitations.

Response: Thank you for the suggestion. I have revised the Discussion section to provide a clearer summary of the key results in the first paragraph. Additionally, I have refined the limitations section to make the content more comprehensive and detailed.

---

## [Decision Letter · Decision Letter 1]

25 Feb 2025

PONE-D-24-42923R1Association between neutrophil count and the risk of cardiovascular disease: A community-based cohort study in TaiwanPLOS ONE

Dear Dr. Yeh,

Thank you for submitting your manuscript to PLOS ONE. After careful consideration, we feel that it has merit but does not fully meet PLOS ONE’s publication criteria as it currently stands. Therefore, we invite you to submit a revised version of the manuscript that addresses the points raised during the review process.

We look forward to receiving your revised manuscript.

Kind regards,

Aleksandra Klisic

Academic Editor

PLOS ONE

Journal Requirements:

Reviewers' comments:

Reviewer's Responses to Questions

**Comments to the Author**

1. If the authors have adequately addressed your comments raised in a previous round of review and you feel that this manuscript is now acceptable for publication, you may indicate that here to bypass the “Comments to the Author” section, enter your conflict of interest statement in the “Confidential to Editor” section, and submit your "Accept" recommendation.

Reviewer #1: All comments have been addressed

Reviewer #2: All comments have been addressed

Reviewer #3: All comments have been addressed

2. Is the manuscript technically sound, and do the data support the conclusions?

Reviewer #1: Yes

Reviewer #2: Partly

Reviewer #3: Yes

3. Has the statistical analysis been performed appropriately and rigorously? 

Reviewer #1: Yes

Reviewer #2: Yes

Reviewer #3: Yes

4. Have the authors made all data underlying the findings in their manuscript fully available?

Reviewer #1: (No Response)

Reviewer #2: Yes

Reviewer #3: Yes

5. Is the manuscript presented in an intelligible fashion and written in standard English?

Reviewer #1: Yes

Reviewer #2: Yes

Reviewer #3: Yes

6. Review Comments to the Author

Reviewer #1: I commend the authors for their valuable study and their precise response to the provided comments. I have no further question

Reviewer #2: Hello. This is a good article and the topic is explained correctly. Thanks for answering the questions of the previous version.

Good luck.

Reviewer #3: REVIEWER COMMENT AND SUGGESTIONS

Overall, congratulations to the authors for resubmitting this manuscript. However, there are a few minor issues that need to be addressed.

Tittle: The title is quite explicit, but I observed that in line 4, there is no need to rewrite the title.

ABSTRACT

Background: it is unnecessary to include the design on a portion of the background; instead, you should condense the problem statement and the objective of your research..

Result: In the results section, the authors should begin with the findings instead of starting with demographic information.

Conclusion: there is no need to write the design of the study the authors should conclude according to the result.

Methods: The authors need to make revisions to this section, specifically line number 129.

Result: I observed that line number 198 is unnecessary, so please eliminate it

7. PLOS authors have the option to publish the peer review history of their article (what does this mean? ). If published, this will include your full peer review and any attached files.

**Do you want your identity to be public for this peer review?** For information about this choice, including consent withdrawal, please see our Privacy Policy .

Reviewer #1: **Yes: ** Mirsaeed Abdollahi

Reviewer #2: No

Reviewer #3: **Yes: ** rehema abdallah

---

## [Author Response · Author response to Decision Letter 2]

11 Mar 2025

Response to Reviewer #3

Dear Reviewer,

Thank you for your helpful comments and suggestions. They have helped us improve the clarity and overall quality of our manuscript. Below, we provide our responses to each of your points and explain the revisions.

1. Reviewer Comment:

Tittle: The title is quite explicit, but I observed that in line 4, there is no need to rewrite the title.

Response: Thank you for your suggestion. The text in line 4 is the short title, which is one of the requirements of PLOS ONE to help readers quickly understand the key points of the study's topic.

2. Abstract

Reviewer Comment:

A. Background: it is unnecessary to include the design on a portion of the background; instead, you should condense the problem statement and the objective of your research.

Response: Thank you for your suggestion. I have revised the Background section of the abstract to make it more concise and focused on the research problem and objective. Below is the revised paragraph:

"Neutrophil count is associated with atherosclerotic plaque formation and cardiovascular diseases (CVD). As previous studies have been predominantly conducted in Caucasians, the significance of neutrophil count as a clinical factor in CVD among other ethnic groups remains unclear."

B. Result: In the results section, the authors should begin with the findings instead of starting with demographic information.

Response: Thank you for your suggestion. I have revised the Results section so that it starts with the study’s main findings instead of demographic details. Below is the revised paragraph:

“Over a median follow-up period of 22 years, 400 cases of new-onset CVD were recorded. Cox proportional hazards regression analysis revealed that a higher neutrophil count was independently associated with CVD incidence in Taiwanese adults, with an HR of 1.42 (95% CI 1.03–1.94) after adjusting for multiple covariates. This association remained consistent in both the subgroup and sensitivity analyses.”

C. Conclusion: there is no need to write the design of the study the authors should conclude according to the result.

Response: Thank you for your suggestion. I have revised the conclusion so that it focuses on the study’s findings. Below is the revised paragraph:

“Our study demonstrated that, in the Taiwanese population, a higher neutrophil count was associated with a higher incidence of CVD over an average 22-year follow-up in individuals without preexisting CVD.”

3. Reviewer Comment:

Methods: The authors need to make revisions to this section, specifically line number 129.

Response: Thank you for your suggestion. I have revised line 124 (new location) and checked the Methods section to improve clarity. Below is the revised paragraph:

“The endpoint of the study was the incidence of cardiovascular events ascertained from hospital records or death certificates, and participants were followed-up until the end of 2013. Cardiovascular events included those related to CAD and CVA. The definition of CAD included fatal and nonfatal CADs, confirmed through death certificate hospitalization records of conditions that required coronary artery bypass grafting (CABG) or coronary angioplasty, respectively. The definition of CVA included neurological symptoms of vascular origin lasting more than 24 hours, resulting in hospitalization or death. CVA subtypes, including hemorrhagic and ischemic strokes, were diagnosed using imaging; however, patients with transient ischemic attacks (TIAs) were excluded. For individuals who experienced both types of events, the endpoint was defined as the time of the first event.”

4. Reviewer Comment:

Result: I observed that line number 198 is unnecessary, so please eliminate it

Response: Thank you for your suggestion. Line 198 refers to the abbreviation for standard deviation (SD). I have removed this sentence. Please let me know if there are any other modifications needed.

---

## [Decision Letter · Decision Letter 2]

25 Mar 2025

Association between neutrophil count and the risk of cardiovascular disease: A community-based cohort study in Taiwan

PONE-D-24-42923R2

Dear Dr. Yeh,

We’re pleased to inform you that your manuscript has been judged scientifically suitable for publication and will be formally accepted for publication once it meets all outstanding technical requirements.

Kind regards,

Aleksandra Klisic

Academic Editor

PLOS ONE

Additional Editor Comments (optional):

Reviewers' comments:

Reviewer's Responses to Questions

**Comments to the Author**

1. If the authors have adequately addressed your comments raised in a previous round of review and you feel that this manuscript is now acceptable for publication, you may indicate that here to bypass the “Comments to the Author” section, enter your conflict of interest statement in the “Confidential to Editor” section, and submit your "Accept" recommendation.

Reviewer #3: All comments have been addressed

2. Is the manuscript technically sound, and do the data support the conclusions?

Reviewer #3: Yes

3. Has the statistical analysis been performed appropriately and rigorously? 

Reviewer #3: Yes

4. Have the authors made all data underlying the findings in their manuscript fully available?

Reviewer #3: Yes

5. Is the manuscript presented in an intelligible fashion and written in standard English?

Reviewer #3: Yes

6. Review Comments to the Author

Reviewer #3: Reviewer comment and suggestions

Overall, the manuscript offers a well-organized investigation into the correlation between neutrophil count and cardiovascular disease (CVD) within a Taiwanese demographic, filling an important void in current literature that mainly examines Caucasian populations. Nonetheless, there are a few specific aspects that require attention.

Strengths:

1. Defined Purpose: The research effectively communicates its aim of exploring the importance of neutrophil count within a non-Caucasian demographic, which significantly enhances the existing literature.

2. Thorough Methodology: The inclusion of a large participant base (2,955 individuals) and an extensive follow-up duration (22 years) creates a strong dataset that bolsters the credibility of the findings.

3. Statistical Assessment: The application of Cox regression models appropriately accounts for potential confounding variables and facilitates a detailed understanding of the link between neutrophil count and cardiovascular disease risk.

4. Subgroup and Sensitivity Evaluations: These assessments contribute integrity and help confirm the reliability of the outcomes across various demographics and data ranges.

Areas for Improvement:

1. Methods Elaboration: The abstract states that Cox regression models were utilized, but it would be helpful to include a brief overview of the covariates considered in the analysis to shed light on the range of potential confounding factors examined.

2. Limitations of the Study: The study does not address any limitations associated with the study, such as possible biases that may arise from observational study designs or the specific criteria employed to identify new-onset CVD. Recognizing these would help provide a more comprehensive understanding of the results.

3. Consistency in Terminology: It is important to maintain consistent terminology, such as "neutrophil count" and "CVD." Any additional terms introduced in the full manuscript should be clearly defined upon their first appearance.

4. Conclusions and Suggestions: While conclusions are made, the study could expand on the practical implications of the findings for clinical practice and future research, indicating how these results could affect healthcare strategies within the Taiwanese population and elsewhere.

Weaknesses:

Lack of Specificity in Methods: The abstract does not provide sufficient detail on the covariates considered in the Cox regression analysis. Revise and improve

7. PLOS authors have the option to publish the peer review history of their article (what does this mean? ). If published, this will include your full peer review and any attached files.

**Do you want your identity to be public for this peer review?** For information about this choice, including consent withdrawal, please see our Privacy Policy .

Reviewer #3: **Yes: ** Rehema Abdallah

---

## [Editor Report · Acceptance letter]

PONE-D-24-42923R2

PLOS ONE

Dear Dr. Yeh,

I'm pleased to inform you that your manuscript has been deemed suitable for publication in PLOS ONE. Congratulations! Your manuscript is now being handed over to our production team.

Kind regards,

on behalf of

Dr. Aleksandra Klisic

Academic Editor

PLOS ONE